DNA barcoding unveils skate (Chondrichthyes: Rajidae) species diversity in ‘ray’ products sold across Ireland and the UK

Griffiths Andrew Mark 1 andiff100@googlemail.com
Miller Dana D. 2 3
Egan Aaron 2
Fox Jennifer 2
Greenfield Adam 1
Mariani Stefano 1
1 School of Environment and Life Sciences, University of Salford , Greater Manchester , UK
2 School of Biology & Environmental Science, Science Education and Research Centre-West, University College Dublin , Belfield, Dublin , Ireland
3 Fisheries Centre, University of British Columbia , Vancouver, British Columbia , Canada
de Vere Natasha
Electronic publication date: 2013 Aug 13
Publication date: 2013
Volume: 1
Electronic Location ID: e129
Received 2013 Apr 21; Accepted 2013 Jul 21
Copyright: © 2013 Griffiths et al.
Copyright year: 2013
Copyright holder: Griffiths et al.
License: This is an open access article distributed under the terms of the Creative Commons Attribution License, which permits unrestricted use, distribution, and reproduction in any medium, provided the original author and source are credited.
License URL: https://creativecommons.org/licenses/by/3.0/

Keywords: Elasmobranch, Species identification, Forensically informative nucleotide sequencing, Mislabelling, DNA barcoding, Skate, Ray

Funding: European Union INTERREG Atlantic Area Program ‘LabelFish’: project 2011-1/163 Irish Research Council (IRC) University College Dublin University of Salford This work was jointly funded by the European Union INTERREG Atlantic Area Program (‘LabelFish’, project 2011-1/163) and the Irish Research Council (IRC). Additional support also originated from The Department for Environment, Food and Rural Affairs (DEFRA), University College Dublin and the University of Salford. The funders had no role in study design, data collection and analysis, decision to publish, or preparation of the manuscript.

==============================
Skates are widely consumed across the globe, but many large species are subject to considerable concern regarding their conservation and management. Within Europe such issues have recently driven policy changes so that, for the first time, reports of skate landings now have to be made under species-specific names. Total allowable catches have also been established for many groups, which have been set to zero for a number of the most vulnerable species (e.g., Dipturus batis, Raja undulata and Rostoraja alba). Whilst accurate species identification has become an important issue for landings, the sale of skates is still usually made under a blanket term of “skate” or “ray”. The matter of identifying species of skate is further complicated by their morphologically conservative nature and the fact that they are commercially valued for their wings. Thus, before sale their bodies are usually discarded (i.e., “winged”) and often skinned, making morphological identification impossible. For the first time, DNA barcoding (of the mitochondrial COI gene) was applied to samples of skate wings from retail outlets across the British Isles, providing insight into which species are sold for consumption. A total of 98 wing samples were analysed, revealing that six species were sold; blonde ray (Raja brachyura), spotted ray (Raja montagui), thornback ray (Raja clavata), cuckoo ray (Leucoraja naevus) small-eyed ray (Raja microocellata) and shagreen ray (Leucoraja fullonica). Statistical testing demonstrated that there were significant differences in the species sold in the distinct retail groups which suggests complex drivers behind the patterns of sale in skates. The results also indicate that endangered species are not commonly being passed on to consumers. In addition, the practice of selling skate wings under ambiguous labels is highlighted as it makes it extremely difficult for consumers to exercise a right to avoid species of conservation concern. Interestingly, a single retailer chain labelled their wings as originating from three smaller-growing species (generally to be considered of lower conservation concern); of the six samples analysed from this company a third were mislabelled and originated from the thornback ray (a larger species that is currently undergoing population declines).

Introduction

Skates (batoid elasmobranchs of the family Rajidae, but colloquially referred to as rays) are consumed across the world with recent estimates of global total catch reaching approximately 720,000 tonnes per annum (Seafish, 2013). Whilst skates are rarely specifically targeted by fisheries, they form a significant by-catch and constitute an important commercial group for human consumption. The group also contains many species that are potentially vulnerable to even low levels of harvesting due to their large size, slow growth, late maturity and low fecundity (Dulvy & Reynolds, 2002). This has famously been demonstrated in the common skate species complex (Dipturus batis), which was the first fish to become locally extinct in parts of its former range through a clear link to commercial fishing (Brander, 1981). Of the 75 rajids assessed by the International Union for the Conservation of Nature (IUCN) red list of threatened species (excluding those defined as data deficient), 19 are listed as threatened (IUCN, 2012). In response to skate declines, the European Union (EU) has now established total allowable catches (TAC) for skates in most European waters, and currently it is ‘prohibited for EU vessels to fish for, to retain on board or to land’ three species; Raja undulata, Raja alba and the D. batis-complex (CEC, 2013).

Until recently, landings of different skate species were combined under a single category, such that serious declines in one species could effectively be masked by stable trends in more abundant groups (Dulvy et al., 2000). Therefore, to more easily assess trends in abundance of individual species, a recent EU Regulation (CEC, 2013) has been introduced to ensure that landings of many skates are recorded by species and not as an aggregated group. Identification remains problematic as skates are a morphologically conservative group, making the process of distinguishing between some species difficult. Furthermore, similarly to shark finning, skates are only commercially valued for their large pectoral fins (or ‘wings’), so prior to sale their bodies are usually discarded (i.e., “winged”) and often skinned, making morphological identification impossible. Molecular genetic techniques for species identification therefore represent the only methods for distinguishing between different skates, allowing for the protection of threatened species or populations, and also the recognition of groups with different commercial values. Indeed, previous studies utilising molecular approaches for species identification of skate species have proved highly effective, often helping to uncover previously cryptic diversity (Díaz de Astarloa et al., 2008; Smith et al., 2008; Griffiths et al., 2010).

In recent years international efforts have been made to establish a standardised reference DNA barcode of life database (BOLD; www.barcodinglife.org; Ratnasingham & Hebert, 2007), with the aim of generating cytochrome-c oxidase I or COI gene sequences to facilitate the identification of all animals. This includes a campaign to DNA barcode all fishes; FISH-BOL (Ward, Hanner & Hebert, 2009). In fact, specific efforts have already been made to collect validated barcode sequences for skates both within Europe (Serra-Pereira et al., 2010), and globally (Ward et al., 2008; Coulson et al., 2011; Lago, Vietites & Espineira, 2012). These reference sequences have now facilitated numerous molecular studies of species identification and mislabelling, which have helped identify high levels of substitution in many commercial products (Marko et al., 2004; Wong & Hanner, 2008; Cawthorn, Steinman & Witthuhn, 2012). Particular insights have also been gained through applications of DNA barcoding to elasmobranch products, which have demonstrated high levels of fraudulent substitutions in sales of sharks (Barbuto et al., 2010), identified endangered species in confiscated shark fins (Holmes, Steinke & Ward, 2009) and revealed the sale of endangered sawfish as “shark” fillets in markets in Brazil (Melo Palmeira et al., 2013).

Despite EU legislation regarding the correspondence of species/Latin names for landings, skates are still generally sold under the broad title of ray or skate wings (with skate wings typically originating from the larger fish). Therefore, little remains known about exactly which species are passed on to consumers. Beside the increasing number of studies reporting substantial levels of seafood mislabelling (Logan et al., 2008; Miller & Mariani, 2010), the importance of guaranteeing informed consumer choice has also been highlighted (Miller, Jessel & Mariani, 2012), which is particularly relevant to the sales of skate. Moreover, in recent years several large supermarket chains have consciously moved from stocking ‘skate’ to ‘ray’ wings, presumably reflecting a desire to sell smaller species from more sustainable sources (Waitrose, 2009). This study represents the first intensive investigation into the patterns of the sale of skate species in the North-east Atlantic, specifically the Republic of Ireland (ROI) and the United Kingdom (UK), with a particular focus on (a) whether those species that are prohibited to land within the EU appear on sale, and (b) if market variations can be detected between different retail sectors and/or regions.

Materials & Methods

Sample collection

Samples labelled as ray or skate wings were initially purchased from fishmongers or fish counters within supermarkets and takeaway ‘fish and chips’ shops from across Dublin (ROI). The retailers were also visited in two distinct periods; January–February and October–November 2010. A total of 58 wings from 10 fish mongers/counters and 9 takeaways were analysed (Table 1 and Supplemental Information). In order to provide a wider view of patterns of skate consumption, 40 additional fresh ray or skate wings were also collected between October 2012 and March 2013 from around the UK; sampling was focused in the south-west of England (i.e., counties of Devon and Cornwall), north-west England (largely in Greater Manchester and Liverpool), Glasgow (Scotland), and Cardiff (Wales). UK samples were limited to ‘fresh’ products as investigations of takeaways in the UK cities failed to identify any that sold skate. Repeat sampling from any single shop was kept to a minimum, although it was necessary in some cases to increase the sample size (but efforts were generally made to sample on different dates or from different sized wings; Supplemental Information 1). Therefore, between January 2010 and March 2013 a total of 98 wings were collected from 49 different retailers across the ROI and UK.

Table 1 Details of ray wing sampling, summarising the numbers of retailers, numbers of samples and dates of collection at each location.

Further information is given in the Supplemental Information 1.

Location	Country	Retailer	No. retailers	No. samples	Date	
Dublin	ROI	Monger	10	18	Jan 2010–Feb 2010	
Dublin	ROI	Take-away	9	16	Jan 2010–Feb 2010	
Dublin	ROI	Monger	9	13	Oct 2010–Nov 2010	
Dublin	ROI	Take-away	8	11	Oct 2010–Nov 2010	
NW England	UK	Monger	9	11	Nov 2012–Feb 2013	
SW England	UK	Monger	11	11	Oct 2012–Mar 2013	
Glasgow	UK	Monger	3	9	Jan 2013	
Cardiff	UK	Monger	7	9	Jan 2013	

Molecular analysis

Tissue samples were collected following Miller, Jessel & Mariani (2012) and DNA extractions were completed according to a modified chloroform extraction procedure (Petit, Excoffier & Mayer, 1999). DNA barcoding (specifically of COI gene) was selected as the species identification tool (Hebert et al., 2003). Approximately 648 base pairs (bp) of the COI region were amplified by polymerase chain reaction (PCR) using primers designed by Ward et al. (2005): FishF2 (5′-TCGACTAATCATAAAGATATCGGCAC-3′) and FishR2 (5′-ACTTCAGGGTGACCGAAGAATCAGAA-3′). Reaction conditions following Serra-Pereira et al. (2010) were utilized for all purchased skate samples. The PCR products were sequenced by Macrogen Inc, Seoul, and Source Bioscience, Nottingham, and the results were checked by eye in BIOEDIT version 7.1.11 (Hall, 1999) for errors and ambiguous base calls. The sequences were then input into the BOLD system using the species level barcode database to identify the species of each sample and then cross-referenced using BLAST on GenBank (Basic Local Alignment Search Tool, National Centre for Biotechnology Information, Bethesda, Maryland; www.ncbi.nlm.nih.gov/). A 98% similarity criterion was used as a threshold above which identification of unknown samples was deemed reliable (as analysis of barcode records suggests at a divergence of greater than 2% it is very likely that sequences belong to different species, Ward, 2009). Any additional matches within this 98% criterion are also reported in the Supplemental Information. However, in the case of the BOLD system, only additional matches within the Public Record Barcode database are described. This should help ensure that the primary identification is made with a comprehensive database that maximises the chance of finding a strong match, but any additional matches are only made with the more closely regulated dataset, helping to reduce issues associated with the misidentification of submitted records.

Data analysis

Principal component analysis (PCA) of the data was conducted in PRIMER-6 (Clarke & Warwick, 2001), with each retailer representing an individual data point in the ordination. The software was also used to conduct a non-parametric analysis of similarity (ANOSIM) with 999 permutations, and to calculate the contribution of variables to similarity (or similarity percentages; SIMPER). Both utilised the Bray-Curtis similarity measure during calculation. Otherwise, the program defaults were utilised.

Results & Discussion

DNA barcoding

Good quality sequences were obtained for all 98 skate samples, these varied in length between 494 and 646 bp (average length 617.9 bp). Searches on BOLD and GenBank generally produced clear matches allowing for confident assignment of species and there was good agreement between databases. In fact, all the searches yielded matches that were within 98% similarity to database records, with one exception (Supplemental Information 1). In this case a 100% match on BOLD was made, but no results within the 98% similarity criterion were located in GenBank, so the BOLD identification was utilised. Additionally, in 24 cases a second close match (also with less than 2% divergence to the unknown samples) was also identified. These additional matches originated from closely related sister taxa, but in every case the species were not distributed around the British Isles (they included: Raja maderensis that is restricted to Madeira and the Azores, Raja polystigma that is restricted to the Mediterranean, Raja straeleni and Leucoraja walacei that both have a primarily South African distribution). Therefore, the additional matches were deemed unlikely to be the true source of the samples and the species with the highest degree of similarity from the BOLD species level barcode database was upheld for the identification.

Across the entire study, six species of skate were identified in products sold for consumption (Table 2); 31 blonde rays (Raja brachyura), 28 spotted rays (Raja montagui), 19 thornback rays (Raja clavata), 14 cuckoo rays (Leucoraja naevus), five small-eyed rays (Raja microocellata) and one shagreen ray (Leucoraja fullonica).

Table 2 Frequency of the skate species identified in each of the sample collections, across the 98 samples analysed.

Location	Country	Retailer	Date	Thornback	Blonde	Spotted	Small-eyed	Cuckoo	Shagreen	
Dublin	ROI	Monger	Jan 2010–Feb 2010	5	3	9	0	1	0	
Dublin	ROI	Take-away	Jan 2010–Feb 2010	6	8	1	1	0	0	
Dublin	ROI	Monger	Oct 2010–Nov 2012	1	5	5	0	2	0	
Dublin	ROI	Take-away	Oct 2010–Nov 2012	2	9	0	0	0	0	
Various	UK	Monger	Oct 2012–Mar 2013	5	6	13	4	11	1	
			TOTAL	19	31	28	5	14	1	

Statistical support

The PCA failed to produce highly distinct clusters based on retail type or region, although it does broadly illustrate some important patterns that are evident within the data (Fig. 1, because tests detailed below failed to identify significant differences between samples collected in different seasons, the samples from individual retailers were combined across months). Interpretation of the PCA axis and eigenvectors (Fig. 1 and Supplemental Information 2) suggests blonde rays are the main constituent in sales from takeaways, whilst spotted rays are particularly important in fishmongers, leading to a degree of separation of retailer type in PC1. The PCA also illustrates how cuckoo, shagreen and small-eyed rays are largely absent in samples analysed from the ROI, occurring much more commonly in the analysis of UK ray wings.

Figure 1 Principal components analysis of retailers based on species of skates sold (principal component 1 explains 40.2% and principal component 2 explains 28.4% of the observed variation).

The result of the ANOSIM when testing for a difference between the temporally replicated samples collected in January/February and October/November 2010 from Dublin was non-significant (R = 0.073, p = 0.052). As the result was close to the 95% confidence interval, tests were also made between the temporal replicates of the fishmongers (R = 0.091, p = 0.130) and takeaways (R = 0.017, p = 0.298) separately, which were also non-significant. In contrast, the comparison between Dublin fishmongers and takeaways was highly significant (R = 0.182, p = 0.001), with the SIMPER results demonstrating average dissimilarity of 69.89% between samples (with the highest contributions originating from the blonde and spotted rays, each contributing about a third of the total dissimilarity). Comparison of fishmongers in the ROI and the UK was non-significant (R = −0.12, p = 0.995).

Conservation issues

Across the study, no species with a vulnerable status on the redlist (IUCN, 2012), or an EU zero TAC, was identified. Given the severe declines in abundance of these skate species, this may simply reflect the relatively modest sample collection assembled here, but considerable efforts across the British Isles have been made to emphasise the conservation status and TAC limits placed on many skates (e.g., Seafish, 2013). Therefore, the complete absence of species of critical conservation concern may also reflect effective controls by the EU and efforts of fishermen and stakeholders to reduce landings of these groups. Perhaps a more stringent test of such action could be made in regions where threatened species are more locally abundant: e.g., R. undulata remains relatively common around the Channel Isles (Ellis, McCully & Brown, 2012), and analyses focused on this area could form a more rigorous test of effective landing controls. Interestingly, the largest growing species identified (R. brachyura) also occurred most frequently in the study. Maybe more awareness should focus on R. brachyura as size has been suggested as a good proxy for vulnerability to overexploitation (Dulvy & Reynolds, 2002). R. brachyura, R. clavata and L. fullonica that were identified in the purchased wings are already included under the ‘near-threatened’ category in the IUCN redlist. R. brachyura is also listed under the red category in the Marine Conservation Society’s good fish guide (MCS, 2013), which suggests it should not be so extensively consumed. Perhaps guidelines on the maximum size of skate wings sold would increase consumer confidence that the products originate from sustainable sources and reinforce action to protect some of the vulnerable large growing species. This would certainly simplify consumer choice, but could potentially overlook trading or harvesting of juveniles from the large growing species that could go undetected due to their small size.

Variation in patterns of sale

This study represents a relatively intensive effort to collect samples from ray wings across multiple time-points, regions and countries, and goes beyond many previous investigations of seafood and mislabelling that focused on a single city, port or landing. Notwithstanding these efforts, it has resulted in a relatively modest sample collection. In part this reflects the fact that ray wings were not commonly available in some areas (particularly within the UK, where many retailers were visited that did not stock them), but it also reflects a desire to avoid high levels of repeat sampling from any one location or supermarket chain. It is hoped this approach gives the samples analysed a high degree of independence (not just repeatedly sampling from the same supply chains), producing a truly comprehensive examination of the sale of skates. Therefore despite the modest sample size, this study represents the first occasion that an investigation of the patterns of sale of ray wings can be made between retailer types, seasons and regions.

A highly significant difference was identified in species sold between fishmongers and takeaways within Dublin. The additional level of processing in takeaways provides greater opportunity for large wings originating from larger species to be divided before sale. Conversely, wings from smaller growing species may form more attractive sizes for sale in fishmongers. In recent years a number of supermarkets have also made a move towards selling wings from smaller skate species (Waitrose, 2009), marketing them as sustainably sourced ‘ray’ wings. As 28 of the ray wings sampled across the entire study in the fishmonger group originated from supermarket fish counters, this may help explain the difference in species sold. It is also interesting to note that whilst all the samples analysed were labelled as “ray” or “skate” (and all the species identified belonged to the Rajidae family, complying with broad expectations of the labels), one supermarket chain went further and specified its wings originated from three smaller growing species. In the six cases where wings were analysed from this chain two belonged to R. clavata that was not included on the label, therefore, demonstrating a 33% level of mislabelling within this small subset of samples. Overall, the sale of skates under ambiguous names remains a challenge to the consumers’ ability to exercise a right to choose those from less vulnerable groups.

A somewhat surprising result from the data analysis was the failure to identify any significant difference in the patterns of species sold in the January/February and October/November sample collections from Dublin. Many elasmobranch species are highly migratory, particularly with reference to specific nursery areas, with marked seasonal movements between the North Sea and Thames estuary for R. clavata having already been observed (Hunter et al., 2005). Therefore, it may be expected that such seasonal movements would affect capture vulnerability in commercial fisheries, and hence patterns of availability to consumers. One explanation for this result is that retailers will freeze stocks of fish to ensure a steady supply. In the case of skates, previous work suggests this is not common, even within takeaway shops, as the wings do not freeze well so supply was dependent on local fisheries (Miller & Mariani, 2013). However, much remains unknown about the exact provenance of samples sold, behaviour of fisherman and even the movements of skate species (Ellis et al., 2005), such that interpreting patterns of sale of skates remains difficult.

The same problems for interpretation occur when comparing sales of skate species in the different countries (ROI and UK), which were also non-significant. Perhaps this reflects the fact that the sale of ray wings in both countries exploits similar species and stocks of skates. Although in this case the comparison is also confounded by the fact that sample collection was conducted approximately two years apart. The only other study to utilise genetic methods of species identification on skate products was recently conducted by Lago, Vietites & Espineira (2012) in Spain. The 10 samples of whole or winged fresh skate analysed there originated not only from species found around Spain, but included groups restricted to other areas of Europe and even South America. This suggests patterns of sale could vary dramatically between countries or regions and they will not simply reflect levels of abundance in local fisheries.

Conclusions

This study represents the first intensive effort to investigate patterns in the sale of skate wings. The results show that no species with a vulnerable status on the redlist (IUCN, 2012), or with prohibited landings in the EU, appear to contribute to the market. Given the low abundance of many of these species further investigations may be required to accurately assess whether these groups may reach points of sale anywhere in Europe. Investigations at the point of landing, and throughout the supply chain, would also provide interesting insights into accuracy of identification and reveal if any species are landed, but withdrawn, before reaching retailers. The results also demonstrate significant differences between the retailers (fishmonger and takeaway) included in the study. Understanding why fishmongers also tended to sell smaller-growing species is challenging, but may relate to a desire to supply less vulnerable groups. This study also highlights that the use of ambiguous and amalgamated sales terms makes it extremely difficult for consumers to exercise a right to choose species of lesser conservation concern. Interestingly, analysis of six wings from the only retailer to label its ray wings as originating from three smaller growing species showed two (33%) actually originated from R. clavata, a larger species with a decreasing population trend that is listed as near threatened on the IUCN redlist (IUCN, 2012).

Supplemental Information

Supplemental Information 1 Details of the samples collected (location and date of purchase), length of DNA barcode amplified, BOLD sample ID and the results of searches on BOLD and GeneBank databases

Click here for additional data file.

Supplemental Information 2 Summary of the Principal Component Analysis (PCA) detailing eigenvalues and eigenvectors

Click here for additional data file.

Carlotta Sacchi, Jennifer Coughlan and Belgees Boufana are gratefully acknowledged for their technical support during this project. Thanks are also due to Phil Cannon, Mary Lane, Richard Cooper and Sam Cooper for their aid in supporting sample collection.

Additional Information and Declarations

Competing Interests

Author Contributions

DNA Deposition

The authors have no competing interests to declare.

Andrew Mark Griffiths conceived and designed the experiments, performed the experiments, analyzed the data, contributed reagents/materials/analysis tools, wrote the paper.

Dana D. Miller performed the experiments, contributed reagents/materials/analysis tools, wrote the paper.

Aaron Egan, Jennifer Fox and Adam Greenfield performed the experiments.

Stefano Mariani conceived and designed the experiments, contributed reagents/materials/analysis tools, wrote the paper.

The following information was supplied regarding the deposition of DNA sequences:

BOLD - all sequences have been entered into the database and the sample IDs are included in Supplemental Information 1.

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
