# Peer review of "DNA barcoding unveils skate (Chondrichthyes: Rajidae) species diversity in ‘ray’ products sold across Ireland and the UK"

_PeerJ, doi:10.7717/peerj.129_

## Round 0.1 · original submission · Major Revisions

All three reviewers agree that this paper is interesting, relevant and potentially suitable for publication in PeerJ. Reviewer 2 suggests a small number of changes whilst reviewers 1 and 3 suggest some more substantial changes that will help to improve the manuscript.

The major areas for modification include.

The sample size is relatively small but is acceptable but the statistical comparisons currently used are not all appropriate to such small sample sizes. Some of the comparisons also have confounding variables. For example, given the difference in years between the UK and ROI sampling investigation of a regional effect is not really appropriate. The statistics need simplifying to answer questions appropriate to the sample size.

Reviewers 1 and 3 mention that more explanation is required about the analysis of the molecular data, in particular the methods used for comparing your sequences to BOLD and GenBank. Reviewer 3 would also appreciate more information and justification on the sampling methods used to collect the fish samples.

Reviewer 1 states that the literature cited is not up to date and provides a useful list of supplementary papers.

Reviewer 1 ·

Basic reporting

The article is focused on a methodology, the molecular identification of seafood for monitoring and control of frauds, which is increasingly adopted with success.
It reconfirms the utility of the DNA barcoding approach in identifying seafood products and provides useful information on the identity of fresh ‘wings’ of skates (rays) sold in UK and Republic of Ireland.
However, it suffers of three main limitations:
1 – a very restricted sample size, especially for UK when only 20 specimens have been analysed, to pretend performing sound static analyses and to draw conclusions on the differences between retailers, seasons and countries.
2 – bibliographic references are incomplete, obsolete, and/or often self-referenced. Furthermore, relevant prior literature is not always appropriately referenced.
A few examples (but more can be listed):
- data on global catches and TACs refers to Seafish 2009 but a ‘version 2013’ of the same document exists and it contains updated informations and different numbers; furthermore, in both documents the original source of data for catches is FAO, while not directly citing it?
- the current EC Regulation, specifying TACs in the different EU zones, is EC N°39/2013 the cited EC N° 43/2009 is not anymore in place.
- The citations of previous studies that utilise molecular approaches for species identification are very imcomplete, as well as these of papers dealing with seafood identifation/mislabeling. To give an idea on how the references can be implemented, an inexhaustive list is provided at the end of this section.
- Documents by MCS, Seafish, Waitrose are frequenly cited as source of relevant information, are these data scientifically-based? No other sources are available?
3 - finally, despite the fact that in the abstract the authors claim to discuss about the potential for guidelines on maximum size of skate wings, in the discussion section this aspect is only superficially addressed.

Potential useful references:
Ardura, A., I. G. Pola, et al. (2010). "Application of barcoding to Amazonian commercial fish labelling." Food Research International 43: 1549-1552.
Barbuto, M., A. Galimberti, et al. (2010). "DNA barcoding reveals fraudulent substitutions in shark seafood products: The Italian case of “palombo” (Mustelus spp.)." Food Research International 43: 376-381.
Cawthorn, D.-M., H. A. Steinman, et al. (2012). "DNA barcoding reveals a high incidence of fish species misrepresentation and substitution on the South African market." Food Research International 46: 30-40.
Cerutti-Pereyra, F., M. G. Meekan, et al. (2012). "Identification of rays through DNA barcoding: an application for ecologists." PloS one 7: e36479.
Costa, F. O., M. Landi, et al. (2012). "A Ranking System for Reference Libraries of DNA Barcodes: Application to Marine Fish Species from Portugal." PLoS ONE 7: e35858.
COULSON, M. W., D. DENTI, et al. (2011). "DNA barcoding of Canada’s skates." Molecular Ecology Resources 11: 968-978.
Dudgeon, C. L., D. C. Blower, et al. (2012). "A review of the application of molecular genetics for fisheries management and conservation of sharks and rays." Journal of fish biology 80: 1789-1843.
Filonzi, L., S. Chiesa, et al. (2010). "Molecular barcoding reveals mislabelling of commercial fish products in Italy." Food Research International 43: 1383-1388.
Galimberti, A., F. De Mattia, et al. (2013). "DNA barcoding as a new tool for food traceability." Food Research International 50: 55-63.
Hanner, R., S. Becker, et al. (2011). "FISH-BOL and seafood identification: geographically dispersed case studies reveal systemic market substitution across Canada." Mitochondrial DNA 22 Suppl 1: 106-122.
Holmes, B. H., D. Steinke, et al. (2009). "Identification of shark and ray fins using DNA barcoding." Fisheries Research 95: 280-288.
Keskin, E. and H. H. Atar (2012). "Molecular identification of fish species from surimi-based products labeled as Alaskan pollock." Journal of Applied Ichthyology 28: 811-814.
Kim, S., H.-S. Eo, et al. (2010). "DNA barcode-based molecular identification system for fish species." Molecules and Cells 30: 507-512.
Ko, H.-L., Y.-T. Wang, et al. (2013). "Evaluating the accuracy of morphological identification of larval fishes by applying DNA barcoding." PloS one 8: e53451.
LAKRA, W. S., M. S. VERMA, et al. (2011). "DNA barcoding Indian marine fishes." Molecular Ecology Resources 11: 60-71.
Mabragaña, E., J. M. de Astarloa, et al. (2011). "DNA Barcoding Identifies Argentine Fishes from Marine and Brackish Waters." PLoS ONE 6: e28655.
Marko, P. B., H. a. Nance, et al. (2011). "Genetic detection of mislabeled fish from a certified sustainable fishery." Current biology : CB 21: R621-622.
Melo Palmeira, C. A., L. F. da Silva Rodrigues-Filho, et al. (2013). "Commercialization of a critically endangered species (largetooth sawfish, Pristis perotteti) in fish markets of northern Brazil: Authenticity by DNA analysis." Food Control 34: 249-252.
Moura, T., M. C. Silva, et al. (2008). "Molecular barcoding of north-east Atlantic deep-water sharks: species identification and application to fisheries management and conservation." Marine and Freshwater Research 59(3): 214-223.
Naylor, G. J. P., J. N. Caira, et al. (2012). "A DNA Sequence–Based Approach To the Identification of Shark and Ray Species and Its Implications for Global Elasmobranch Diversity and Parasitology." Bulletin of the American Museum of Natural History: 1-262.
Nicolè, S., E. Negrisolo, et al. (2012). "DNA barcoding as a reliable method for the authenticatio of commercial seafood products." Food technology and biotechnology 50: 387-398.
Rasmussen, R. S. and M. T. Morrissey (2009). "Application of DNA-Based Methods to Identify Fish and Seafood Substitution on the Commercial Market." Comprehensive Reviews in Food Science and Food Safety 8: 118-154.
Ribeiro, A. D. O., R. A. Caires, et al. (2012). "DNA barcodes identify marine fishes of São Paulo State, Brazil." Molecular ecology resources 12: 1012-1020.
Smith, P. J., S. M. McVeagh, et al. (2008). "DNA barcoding for the identification of smoked fish products." Journal of Fish Biology 72: 464-471.
Steinke, D. and R. Hanner (2011). "The FISH-BOL collaborators' protocol." Mitochondrial DNA 22: 10-14.
Steinke, D., T. S. Zemlak, et al. (2009). "DNA barcoding of Pacific Canada’s fishes." Marine Biology 156: 2641-2647.
Steinke, D., T. S. Zemlak, et al. (2009). "Barcoding Nemo: DNA-Based Identifications for the Ornamental Fish Trade." PLoS ONE 4: e6300.
Ward, R. D., R. Hanner, et al. (2009). "The campaign to DNA barcode all fishes, FISH-BOL." Journal of Fish Biology 74: 329-356.
Ward, R. D., B. H. Holmes, et al. (2008). DNA barcoding Australasian chondrichthyans: results and potential uses in conservation. Marine and Freshwater Research. 59: 57-71.
Warner, K., W. Timme, et al. (2013). Oceana study reveals seafood fraud nationwide. Downloaded May/ 23/2013 from http://oceana.org/sites/default/files/reports/National_Seafood_Fraud_Testing_Results_FINAL.pdf
Wong, E. H.-K. and R. H. Hanner (2008). "DNA barcoding detects market substitution in North American seafood." Food Research International 41: 828-837.
Zhang, J. (2011). "Species identification of marine fishes in china with DNA barcoding." Evidence-based complementary and alternative medicine : eCAM 2011: 978253.
Zhang, J.-B. and R. Hanner (2011). "DNA barcoding is a useful tool for the identification of marine fishes from Japan." Biochemical Systematics and Ecology 39: 31-42.

Experimental design

No sampling design exists, the samples seems to have been collected in a random, unplanned chaotic manner and in insufficient numbers for the goals of the paper.
The molecular part (methods) is well described and it is in accordance with the standard procedure for the barcoding of animal species. However, the threshold for accepting as valid the species identification is not specified; a 98% similarity criterion against the entries in databanks was adopted by the authors?, but a least one species with a 95% value was found. Furthermore, the occurrence of samples for which the database searches yielded matches to a second species is reported in Supplementary table 1; these results are not adeguately explained or discussed/commented in the main text.
Even if the valid data come prevalently from the DNA analyses, the molecular part is largely under-represented when compared with the ‘statistical’ one. It is worth expanding and deeping the first one while reducing the second one.
In fact, the statistical part occupies a disproportional part of the paper, also considering that it suffers from the paucity of data (samples). It could be greatly even cancelled, while waiting to collect a more representative number of samples.

Validity of the findings

The results of the PCA analysis are fully described and illustrated in Figure 1 and supplementary Table 2 but not explained at all in the material and method section.
The results & discussion section is lenghty and redundant; in particular the paragraph ‘variation in patterns of sale’, based on the ‘weak’ results of the statistical tests, should be greatly reduced/omissed (see previous paragraph).

Reviewer 2 ·

Basic reporting

Only suggestions are:

Line 70 change ‘This’ to ‘Our’ to make clearer
Line 100 change ‘Herbert’ to ‘Hebert’ (is correct in refs)
Line 130 delete the comma
Line 131 delete ‘the’
Line 152 …. d.f. =3), although, in order…..
Line 177 change ‘were’ to ‘was’

Experimental design

All OK – no comments

Validity of the findings

All OK – no comments (except one)

My very minor comment is that there are chi-square tests readily available using Monte Carlo approaches that are not biased by small expectations. However, the ‘classical’ approach taken is acceptable.

Additional comments

An appropriate use of DNA barcoding methods for a potentially significant consumer issue. Clearly and well written, and an interesting case study.

Reviewer 3 ·

Basic reporting

1.Text from Material and Methods: The whole paragraph referred to BOLDSystems (row 101-107) would be more appropriate in the Introduction. State it in the paragraph referring to Molecular approach for species identification (rows 52-58).
2.References: Need revision in format.
3.Figures and tables: Lack of an accurate legend that clearly explains the contents.

Experimental design

Material and Methods:
1.Authors state that approximately 648bp were amplified (standard length of DNA barcodes), but the same authors open the Results section by affirming that approximately 500bp or more sequences were obtained (row 130). What is the exact range (more could even indicate 510bp...). This length might still be valid for species identification, but the information here provided are not coherent and need further clarification.
2.Sequenced were checked in BioEdit. Checked for what scope? Please specify.
3.Data quality: The authors should be commended for providing thorough details about their work. At the moment, analyses are not reproducible by other scientists. More details about the statistics used need to be given.

Validity of the findings

Results:
1.Why is 98% a valid “threshold” to assess that assignment of species was reliable? Please, refer to Ward R (2009) DNA barcode divergence among species and genera of birds and fishes. Mol Ecol Resour 9: 1077-1085. It would be useful to state and explain the choice of 2% in Material and Methods, and cite the above mentioned paper. This is additionally important also because intra-genus species are treated in the present work.
2.How is the PCA obtained? This analysis is firstly mentioned in the Result section, but it has not been previously described in Material and Methods.
Discussions:
1.Authors claim that “this study represents the first intensive effort to investigate patterns in the sale of skate wings” (row 71 Introduction same as row 272 of Conclusions). “Despite a modest sample collection” is one of the sentences concluding the abstract; “relatively modest sample collection assembled” (rows 178-179). Intensive of modest?
2.Variation in Patterns of Sale: Differences in patterns of species sold in different sampling periods were not significant. Authors explain migratory patterns of the species, and how these might affect capture vulnerability in commercial fisheries and as consequence in what is available in the market. This would work if samples are fresh and collected directly at landings. But samples here analysed are purchased in supermarkets and takeaways from across Dublin. Are those samples fresh, as the 20 collected in UK? Takeaways and supermarkets are likely to purchase big stocks of products, so there is no guarantee on their provenience nor in the real collection time. I personally think that it is not possible to perform this comparison, if the provenience and time of collection are unknown.

Additional comments

The topic addressed in Griffiths et al. is certainly of general interest for the audience of PeerJ as it addresses important issues on species that are of high scientific and management relevance. Information can improve the delineation of fish conservation and fisheries management strategies.
Although it should eventually be published in PeerJ, I think this paper has to be substantially revised and go through another round of reviewing before publication. My main concerns have to do with a few misconceptions, such as like the provenience and real time of collection (particularly non-fresh specimens collected from retailers across Dublin). Studies of these genera should be performed on samples with known origins/fresh samples/landings.

---

## Round 0.2 · accepted · Accept

The authors have made substantial changes to the manuscript taking on board all of the suggestions made by the reviewers or justifying why not within the rebuttal letter. In particular the sample size has been increased allowing more rigorous analysis and conclusions to be made.
This is an important paper highlighting the need for accurate and precise labelling in order to allow consumers to make decisions to choose fish products of lower conservation concern. Of particular interest is that where retailers have attempted to provide more precise information on the species contained within the product this has resulted in mislabelling, illustrating that the supply chain may currently not be very well geared up to allow species level designations to be placed on packaging of ray products.
I look forward to seeing this paper published.
(Points to note at the proof stage: p7 line 165 should be Isles not Iles, p13 line 313 italics needed on Latin binomial)